# Role of Bioactive Compounds in Obesity: Metabolic Mechanism Focused on Inflammation

**DOI:** 10.3390/foods11091232

**Published:** 2022-04-25

**Authors:** Esther Ramírez-Moreno, José Arias-Rico, Reyna Cristina Jiménez-Sánchez, Diego Estrada-Luna, Angélica Saraí Jiménez-Osorio, Quinatzin Yadira Zafra-Rojas, José Alberto Ariza-Ortega, Olga Rocío Flores-Chávez, Lizbeth Morales-Castillejos, Eli Mireya Sandoval-Gallegos

**Affiliations:** 1Academic Area of Nutrition, Interdisciplinary Research Center, Institute of Health Sciences, Circuit Actopan Tilcuautla s/n, Ex hacienda La Concepción, San Agustin Tlaxiaca, Pachuca 42160, Mexico; esther_ramirez@uaeh.edu.mx (E.R.-M.); quinatzin_zafra@uaeh.edu.mx (Q.Y.Z.-R.); jose_ariza@uaeh.edu.mx (J.A.A.-O.); 2Academic Area of Nursing; Institute of Health Sciences, Circuit Actopan Tilcuautla s/n, Ex hacienda La Concepción, San Agustin Tlaxiaca, Pachuca 42160, Mexico; jose_arias@uaeh.edu.mx (J.A.-R.); jimenezs@uaeh.edu.mx (R.C.J.-S.); destrada_luna@uaeh.edu.mx (D.E.-L.); angelica_jimenez@uaeh.edu.mx (A.S.J.-O.); ofloresc@uaeh.edu.mx (O.R.F.-C.); lizbeth_morales@uaeh.edu.mx (L.M.-C.)

**Keywords:** obesity, inflammation, bioactive compounds

## Abstract

Obesity is a disease characterized by an inflammatory process in the adipose tissue due to diverse infiltrated immune cells, an increased secretion of proinflammatory molecules, and a decreased secretion of anti-inflammatory molecules. On the other hand, obesity increases the risk of several diseases, such as cardiovascular diseases, diabetes, and cancer. Their treatment is based on nutritional and pharmacological strategies. However, natural products are currently implemented as complementary and alternative medicine (CAM). Polyphenols and fiber are naturally compounds with potential action to reduce inflammation through several pathways and play an important role in the prevention and treatment of obesity, as well as in other non-communicable diseases. Hence, this review focuses on the recent evidence of the molecular mechanisms of polyphenols and dietary fiber, from Scopus, Science Direct, and PubMed, among others, by using key words and based on recent in vitro and in vivo studies.

## 1. Introduction

Obesity has been cataloged as a slow-motion disaster [1], so exhaustive research to find efficient alternatives against obesity has been the subject of continuous debate. In accordance with the WHO, worldwide obesity has nearly tripled since 1975 (OMS. Global Health Observatory [GHO] data, 2019), and has been considered as a public health problem. Mexico is one of the countries with the highest prevalence of obesity. In addition, obesity is defined as abnormal or excessive fat accumulation that increases the risk of developing a secondary disease. Adipose tissue was previously considered as a static tissue (reservoir for energy). Studies have referred to adipose tissue as a dynamic tissue (metabolically active organ) [2,3,4]. The morphophysiological change of adipose tissue during obesity induces a chronic low-grade inflammatory state, also referred to as parainflammation (intermediate state between basal and inflammatory) or metainflammation (metabolically triggered inflammation) [5,6,7]. On the other hand, visceral adipose tissue could have a local immune response [8], and it is linked with the stimulation and release of detrimental cytokines and chemokines implicated in metabolic disorders [9]. In addition, the inflammation associated with obesity could be triggering other comorbid conditions, such as diabetes, cardiovascular disease, and cancer, among others [2]. 

Some researchers have demonstrated that the consumption of plant-based foods could decline the inflammation state in obesity due to their content of bioactive compounds. Bioactive compounds are found in a minor amount in these food items and they have been reported to be effective in the treatment of obesity [10,11,12,13,14]. Among the compounds reported with beneficial effects are flavonoids, polyphenols, betalains, and fiber, which have been studied as factors with probable effects on specific pathways (PPARα, cyclooxygenase-2, glucose transporter (GLUT4), SIRT1, and PGC1-α) conferring anti-inflammatory activities, with remarkable implications for health and disease [15,16,17] Therefore, this review aims to provide a comprehensive overview of recent studies about the possible role and effect of specific bioactive compounds on weight management and obesity consequences. Several investigations have been focused on the research for natural alternatives that, in agreement with the work, tend to be promising treatments against obesity progression. 

## 2. Physiopathology of Obesity 

Energy self-regulation is a complex system that includes coordinated neurogastrointestinal and endocrine pathways to maintain adequate metabolism and the use of nutrients. Peripheral or afferent systems generate signals exerted on adipocytes (secrete leptin: modulate satiety [18], the pancreas (secretes insulin: regulates the body’s energy supply, cell growth, and metabolism [19]), the stomach (secretes ghrelin: stimulates appetite [20]), and the ileum and colon (secrete peptide YY: appetite regulation [21]). These signals are generated and processed by the arcuate nucleus of the hypothalamus and create new signals that are subsequently emitted by catabolic and anabolic-type neurons. Finally, the efferent system, constituted by hypothalamic neurons, is controlled by the arcuate nucleus and is, therefore, responsible for the effect on food inhibition or intake [22]. Therefore, the effect of food consumption and a lack of caloric expenditure cause obesity through the development of the preadipocyte to a mature state. This event occurs through transcription factors, such as peroxisome proliferator-activated receptor γ (PPARγ), and other transcription factors, including CCAAT/enhancer-binding proteins (C/EBPs, AP-1), signal transducers and activators of transcription (STATs), and Kruppel-like factor (KLF), that promote preadipocyte differentiation into mature adipocytes (adipogenesis) [23]. This mature state is characterized by presenting a low-grade chronic inflammation state caused by the accumulation of proinflammatory macrophages. Immune cells, such as eosinophil, neutrophil, treg cells, and killer T cells, are also responsible for the secretion of inflammatory cytokines, as well as proteins, such as galectin-3, and exosomes [24,25]. On the other hand, the presence of obesity also triggers the activation of nuclear factor erythroid-2-related factor 2 (Nrf2), whose function is characterized by the maintenance of redox and metabolic homeostasis, through regulating the antioxidant endogenous response and decreased inflammatory stress [26,27]. 

### 2.1. Inflammation in Adipose Tissue

Under normal conditions, adipose tissue regulates essential biological processes through the autocrine, paracrine, and endocrine pathways [28,29,30]. When obesity occurs, an inflammatory process originates, which is known as a low–grade chronic inflammation response of prolonged time [31], and is the result of increasing fat tissue (hypertrophy related to an increase in the size of adipocytes and an overproduction of pro-inflammatory mediators by exogenous or endogenous stimuli, Figure 1) due to excess nutrient consumption [32]. 

On the other hand, inflammation of adipose tissue is also described as a body’s natural or biological reaction against pathogens and harmful stimuli caused by toxic compounds, damaged cells, and metabolic factors [33]. There are two types of inflammation: acute (short time), characterized by edema and the migration of leukocytes; and chronic inflammation (long time), differentiated by a constant secretion of proinflammatory molecules by lymphocytes and macrophages on blood vessels and connective tissue [33,34,35,36]. This inflammatory response depends on the origin of the initial stimulus, the location of adipose tissue in the body, as well as the mechanism to counteract it. Existing factors that trigger inflammation, such as (1) cell surface pattern-recognition receptors that detect different detrimental stimuli; (2) the activation of several inflammatory pathways, such as the mitogen-activated protein kinase (MAPK), nuclear factor kappa-B (NF-κB), and Janus kinase (JAK)-signal transducer and activator of transcription (STAT) pathways; (3) the delivery of inflammatory markers as transcription factors: NF-κB, STAT 3, and inflammatory cytokines (TNF-α, IL-1, IL-6, IL-8), pro-inflammatory enzymes as metallopeptidase-9 (MMP-9), cyclooxygenase (COX-2), vascular endothelial growth factor (VEGF), cell adhesion molecules (CAM), such as VCAM-1 and ICAM-1, etc., and (5) the immune response by hypoxia-induced factor (HIF) [33,37,38]. Therefore, factors such as the production of inflammatory intermediaries and dysregulating inflammatory pathways cause the chronic triggering of collateral injury that then impairs tissue homeostasis, developing several chronic diseases related to low-grade inflammation (LGI), such as atherosclerosis, type-2 diabetes, gout, and multiple neurodegenerative diseases, that negatively affect people’s health and life expectancy [17,39,40,41,42].

Many studies report that, during this inflammatory process, there is excessive segregation of inflammatory factors known as adipokines, bioactive molecules responsible for the origin of inflammation and insulin resistance associated with obesity [43], segregated by adipocytes that include TNF-α, IL-6, IFN-γ, plasminogen activator inhibitor (PAI-1), monocyte chemoattractant protein-1 (MCP1), IL-1β, IL-8, IL-10, IL-15, leukemia inhibitory factor (LIF), hepatocyte growth factor (HGF), apolipoprotein amyloid A3 seric (SAA3), macrophage migration inhibitory factor (MIF), potent inflammatory modulators, such as leptin, adiponectin, resistin, and C-reactive protein (CRP), and these maintain both negative and positive effects, such as the maintenance of oxidative stress, changes in autophagy patterns, tissue necrosis, etc. (Table 1).

It has been observed that obesity is related to metabolic pathways, food intake, and energy expenditure (Figure 2), which leads to the alteration of various inflammatory pathways, such as Janus-N-terminal kinase system/signal transducer and transcription activators (JNK/STAT), IκB-kinase β, and protein kinase C (PKC) [44,45], Besides an increased infiltration of cells into adipose tissue [46,47,48,49], due to the systemic circulation of inflammatory factors that stimulate the endothelial cells. Thus, an inflammatory state is triggered by the relationship between adiposity and metabolic pathways, macrophages, adipocytes, and other factors [7,28,50]. All of these alterations trigger the development of other diseases.

**Table 1 foods-11-01232-t001:** Adipokines’ effect on obesity.

Adipokines	Segregation Molecules	Effect	Author
CRP	Increases the expression of vascular cell adhesion molecule-1 (VCAM-1), intracellular adhesion (ICAM-1), and E-selectin in vascular endothelial cells Increases the secretion of monocyte chemoattractant protein-1 (MCP-1)	Participates in the coronary and aortic atherosclerosis that leads to cardiac events	[51]
TNF-α	Decrease of nitric oxide (NO) Increase of endothelin1 (ET-1), angiotensin II (ATII), oxidized low-density lipoproteins (oxLDL), ICAM-1, VCAM-1, MCP-1, CD40/CD40L, and leukocyte adhesion	Increases foam cell formation Increases smooth muscle cell (SMC) proliferation and migration Expansion of the injury area Increases platelet adhesion Increases leptin concentration	[52]
IL-6	Increases the concentration of free fatty acids (FFAs), C-reactive protein (CRP), and nitric oxide (NO)	Induces insulin resistance Decreases hepatic insulin clearance, insulin-dependent hepatic glycogen synthesis, glucose uptake in adipose cells	[53,54]
IL-1β		Inhibition of the insulin-transduction pathway Inhibition of β-cell function Destruction of β-cell mass Induces the transcriptional activation of inflammatory genes	[55,56]
MCP1		Strongly implicated in adipose tissue macrophage (ATM) recruitment, adipose expansion and remodeling, and angiogenesis	[30]
IFN-γ	Cytokine secretion	Induces attraction of monocytes towards the activation of M1-type macrophages originating from proinflammatory cytokine secretion	[57]
PAI-1	Increases the proliferation and migration of smooth muscle cells (SMCs)	Increases foam cell formation Increases platelet adhesion (thrombosis) Inhibition of the residual plasminogen activator	[58]
Resistin	Increases endoteline-1 (ET-1), angiotensine (ATII), oxLDL, intracellular adhesion (ICAM-1), VCAM-1, MCP-1, CD40/CD40L, leukocyte adhesion, and VSMC Stimulates the synthesis and secretion of cytokines in adipocytes and endothelial cells	Decreases NO release Increases in foam cell formation Increases in proliferation and migration of SMC and the expansion of injury area Increases in platelet adhesion and, as a consequence, thrombosis	[59,60]
Visfatin	Induces ICAM-1, VCAM-1, E-selectin, IL-8, IL-6, MCP-1, fibroblast growth factor-2(FGF-2), and metalloproteinase MMP-2/-9 production Increases the release of ROS (reactive oxygen species)		[60,61]
Vaspin		Overexpressed in the obesity state Induces phosphatidylinositol 3-kinase/ Protein kinase (PI3K/AKT) activation, increases both glucose transporter type-4 (GLUT4) expression and translocation, and promotes insulin-stimulated glucose	[62]
Angiotensinogen	Stimulates ICAM, VCAM-1, MCP-1, and factors stimulant of colonies of macrophages M-CSF production.	Decreases NO bioavailability Decreases vasorelaxation mechanisms and increases platelet adhesion to the vascular wall	[44,63]
Leptin	Increases VCAM-1	In hyperleptinemia, the inflammatory process increases Increases oxidative stress Improves vasorelaxation Increases vascular permeability	[52,64]

### 2.2. Obesity and Its Comorbidities

The development of obesity and body mass index (BMI) are concomitant with several chronic diseases, such as type-2 diabetes, cardiometabolic diseases (including hypertension, dyslipidemia, and cardiovascular disease), cancer, non-alcoholic fatty liver disease, among other less prevalent diseases (Table 2) [65,66,67].

#### 2.2.1. Type-2 Diabetes

Since the 1990s, observational studies in humans described that plasma biomarkers of inflammation (CRP and IL.6) are higher in type-2 diabetic patients [68]. Studies in vitro showed that TNF-α could impair insulin signaling in 3T3-L1 adipocytes, leading to the reduced expression of insulin receptor substrate-1 (IRS-1) and Glut4 [69]. In the early 2000s, it was reported that obesity increases low-grade inflammation by maintaining the IKK/NFκB, JNK1/AP1, and PKC pathways [70,71], and this correlates with serum inflammatory markers in type-2 diabetic patients [72,73].

Kahn and collages described immune cell infiltration as concomitant with cytokine secretion by adipose tissue, leading to insulin resistance by retinol-binding protein 4 (RBP4) [74]. They also that found an upregulation of the fatty acid synthesis pathway in the adipose by carbohydrate response element-binding protein (ChREBP), a transcription factor that regulates lipogenesis and glycolysis, leads to GLUT4 overexpression [75].

Chronic obesity progression also induces an inflammatory process in the pancreas caused by the increased flux of no esterified or free fatty acids (FFA) [74] and the subsequent penetration of macrophages to increase cytokine infiltration, including TNF-a, IL-6, and MCP-1 [76], leading to β-cell dysfunction [77]. Likewise, an increase in the glucose and fatty acid levels activates the inflammasome complex in the pancreas, promoting the release of proinflammatory cytokines, such as IL-1β, CRP, IL-6, TNF-α, MCP-1, IL-8, and PAI-1, considered as pro-inflammatory markers in diabetes [43,77,78,79,80,81].

As summarized by Ortega and collages [82], obesity increases chronic inflammation and cytokine production, which affects insulin-dependent tissues and beta cells; peripheral tissues are impaired by the lipotoxicity exerted by ectopic lipid stores in obese subjects, and the increased secretion of a set of autocrine and paracrine products by adiponectin downregulation finally produces the loss of insulin sensitivity concomitant with impaired insulin production in type-2 diabetic patients.

#### 2.2.2. Cardiovascular Disease

Cardiovascular disease (CVD) is one of the first causes of mortality in several countries. Hyperlipidemia, T2D, and hypertension are common pathologies that increase the risk of CVD, and inflammation is a key mechanism for the progression and complications of CVD [83].

Factors that are considered key to the development of endothelial dysfunction are plaque formation and plaque instability, which constitute the main mechanism of vascular damage in atherosclerotic disease. In people with obesity, there is an activation of the systemic inflammation unchained from the accumulation of macrophages in adipose tissue that at the same time stimulate the secretion of pro-inflammatory proteins, mainly TNF-α, IL-6 and C-reactive protein (CRP), leptin, adipocyte fatty acid-binding protein, and several novel adipokines, such as chemerin resistin, visfatin, and vaspin. These inflammatory mediators are responsible for the induction of CVD, such as plaque formation [84], endothelial dysfunction [85], and cardiac dysfunction [86].

In a two-decade prospective follow-up study, the cardiometabolic profile (HOMA-IR, hs-CRP, and serum HDL) was more adverse in recent-onset obesity and persistent obesity youths (23 years old) compared to never obese participants. However, participants who had obesity in early childhood or preadolescence but transitioned to a non-obesity status had similar characteristics to those who were never obese [87]. Therefore, the reduction of weight gain triggers inflammation and cardiometabolic consequences. In a clinical study in adults, the use of an anti-interleukin (IL)-1β antibody in patients with hs-CRP > 2 mg/L decreased the IL-6 and CRP levels associated with decreased cardiovascular events [88].

#### 2.2.3. Cancer

According to different authors, inflammation linked to obesity is considered a risk factor that improves the initiation and progression of various types of cancer [89]. The relationship between obesity and cancer due to alterations such as insulin metabolism, insulin-like growth factor-1 (IGF-1) axis, sex steroids hormones, adipokines, and chronic low-grade inflammation, has been investigated, which contribute to the adverse effects of obesity in cancer development and progression [90,91].

Kolb [92] described that excess nutrients lead to the activation of different metabolic signaling pathways, cytokine release, hyperplasia, and hypertrophy of adipocytes, which, in turn, increased macrophages on white adipose tissue, triggering a low-grade inflammatory response on the organism, promoting a carcinogenic environment.

In addition, the presence of macrophages in obesity causes the infiltration of tumors, and increases the inflammatory tumor microenvironment caused by cytokines, prostaglandins, and angiogenic factors [93]. On the other hand, the obesogenic status also increases growth factor signaling and vascular perturbations, provoking microenvironment changes and inflammation, causing an increased risk of cancer or its progression [93].

#### 2.2.4. Non-Alcoholic Fatty Liver Disease

Non-alcoholic fatty liver disease (NAFLD) is a very complex disorder and is the most common liver disorder related to T2D [94]. NAFLD is characterized by increased lipid accumulation and subsequent inflammatory response to progress to liver cirrhosis, fibrosis, or non-alcoholic steatohepatitis (NASH).

The liver is a metabolic organ that uses fat as fuel during starvation. The increase in ectopic fat and visceral adipose tissue leads to increased secretion of inflammatory markers, such as TNF-α, IL-6, CCL3, soluble intercellular adhesion molecule-1 (sICAM-1), and CRP [95,96].

It has been shown that, during NAFLD, hepatic stellate cells (HSCs) and Kupffer cells increase the secretion of TNF-α and promote the recruitment of immune cells, perpetuating the inflammatory process [97]. On the other hand, the cytokines produced from adipose tissue under obese conditions induce hepatic insulin resistance and fibrosis [98].

**Table 2 foods-11-01232-t002:** Obesity and its relationship with other diseases.

Diseases	Description	Author
Dyslipidemia	This pathology is due to the consequence of lipolysis produced in the adipocyte, increasing the levels of free fatty acids and increasing the synthesis of hepatic triglycerides, which, in turn, leads to an increase in VLDL. On the other hand, the decline in HDL-c is due to the decrease of Apo A-I, CETP, and LCAT, which inhibits the expression of ABCA1, ABCG1, and SR-B1. The cytokines and adipokines are responsible for these alterations in the adipose tissue	[99]
Gallbladder disease	Gallstones originate from the accumulation of cholesterol monohydrate crystals precipitating in gallbladder bile. Therefore, an increase in weight stimulates the risk of gallstones.	[100]
Hyperuricemia	An alteration with increased serum uric acid level development to gout due to monosodium urate crystals depositing mainly in the joints. These conditions increase with obesity due to the production of urates.	[101]
Osteoarthritis	Although the damage is not clear, it has been found that the dysregulation of adipokines (adiponectin, apelin, leptin, lipocalin-2, visfatin, chemerin, and resistin) and cartilage extracellular matrix degradation in the muscle–skeletal system exerts deleterious effects on the joint.	[102]
Hypothyroidism	Lower free irosin 4 and higher tirosin-stimulating Hormone levels are associated with fat accumulation. Modified thyroid function with normal feedback regulation may be the cause of alterations in energy expenditure with subsequent increases in BMI and weight	[103]

## 3. A Brief Overview of the Properties and Dietary Effects of Some Bioactive Compounds in Obesity

Due to the complicated pathophysiology of obesity, it is necessary to strengthen the consumption of a healthy diet based on vegetables to favor a decrease in obesity and its complications. These diets are characterized by the presence of vegetables, grains, or legumes containing single or mixed compounds with synergistic effects [104,105,106,107]. These are named bioactive compounds or phytochemicals [108,109], and are found in all plants as secondary metabolites. The concentration of bioactive compounds (such as fiber dietary, minerals, vitamins, fatty acids, proteins, some carbohydrates, and polyphenols) varies depending on the parts of the plant growth phase and the season [110].

### 3.1. Polyphenol Compounds

Close to 8000 polyphenol compounds have been identified in nature [111], with a great diversity of structures from simple molecules to polymers with high molecular weight [112]. Known as structures of aglycones, the number of aromatic rings depends on the structural elements; polyphenol compounds are classified as flavonoids, phenolic acids, lignans, stilbenes, alkylphenols, curcuminoids, furanocoumarins, phenolic terpenes, and others. Polyphenol compounds are the most abundant phytochemicals in fruit and vegetable-based diets [113,114], and are bioavailable after absorption by the intestine into the circulatory system, after the ingestion of food. However, its bioavailability is affected by various factors, such as food processing, the amount of food ingested, interactions with other molecules, or intestinal factors, which may depend on the pharmacokinetic profile (absorption, distribution, metabolism, and excretion: ADME) [115,116]. Despite this, recent studies on the natural bioactive compounds present in foods linked them with effects on cell functions in obesity, such as a decrease in the inflammatory response [117], inhibit adipogenesis and lipogenesis [118], induce apoptosis [119], regulation genes involved in adipogenesis, lipolysis, and fatty acid oxidation [120], and others.

On the other hand, the biological effect of phenolic compounds on obesity, these compounds maintain other functions, such as the regulation of insult oxidative, inflammation, and autophagy in diabetic nephropathy due to the action mechanism of ferulic acid, which is based on the regulation of the AGE, MAPK, and NF-kB pathways. Additionally, ferulic acid inhibits excessive ROS production, stimulates autophagy, and inhibits apoptotic cell death in a high-glucose environment on cultured NRK-52E cells [121].

Quercetin is effective for gut dysbiosis, improving with the administration of 0.2%, before antibiotic treatment in mice as it restores the diversity of the gut bacteria as well as intestinal barrier function [122]. 

Lignans have been found as a neural protector due to their inhibitory effect on NO production in LPS-activated microglia. Other compound derivative-lignans attenuate the production of NO and PGE2, as well as inhibit the expression of iNOS and COX-2 by suppressing I-kB-a degradation and the nuclear translocation of the p65 subunit of NF-kB [123].

#### 3.1.1. Phenolic Acids

Phenolic acids confer health-promoting properties due to antioxidant functions by the reactivity of the phenol moiety (hydroxyl substituent on the aromatic ring) or electron donation and singlet oxygen quenching [124,125,126]. The study conducted by Aranaz [127] reported that phenolic acids have an inhibitory effect on adipogenesis in 3T3-L1 adipocytes at three different doses (10, 50, and 100 mM) and remained in the medium for 8 days. Additionally, this effect was accompanied by the down-regulation of Scd1 and Lpl, and PPARγ activation by phenolic acid [127]. Another study conducted by Hsu and Yen [128], concluded that 3T3-L1 adipocytes treated with rutin at doses of 0–250 μM for 12 and 24 h have an inhibitory effect on intracellular triglycerides and glycerol-3-phosphate dehydrogenase (GPDH) activity, which could be mediated by a decrease in the expression of adipogenic transcription factors PPAR-γ and C/EBPR, and leptin, as well as an increase in the expression of adiponectin [128]. In general, some studies on phenolic acids have demonstrated the inhibition of macrophage infiltration and inflammatory cytokine release, such as TNFα, MCP-1, and PAI-1 through NF-kB downregulation [129,130,131]. On the other hand, phenolic acids contribute to the increased secretion of anti-inflammatory adiponectin from adipocytes, avoid adipocyte differentiation, and regulate adverse lipid profiles [129]. There is a wide variety of compounds that maintain positive functions against obesity, as shown in Table 2.

#### 3.1.2. Flavonoids

The biochemical activities of flavonoids and their metabolites depend on their chemical structure, which may vary with one or more hydroxyl substituents, including their derivatives. According to different studies, flavonoids have been related to the reduction of weight due to the loss of adipose tissue [132], β-oxidation stimulation [133], adipogenesis, and lipogenesis inhibition by decreasing the expression of LPL, SREBP1c, and PPARγ [134]. In addition, flavonoids (25–100 μM) decrease the mRNA expression of adipogenic transcription factors (*C/EBPá, PPAR*-α, and SREBP-1) on 3T3-L1 cells.

Additionally, it has been reported that flavonoids (quercetin to 10, 50, and 100 μM)) induce the apoptosis of mature adipose tissues through the modulation of extracellular signal-regulate kinase (ERK) ½ and JNK on 3T3-L1 [135]. On the other hand, during the inflammatory response, flavonoids could inhibit the expression and secretion of pro-inflammatory cytokines [136]. Therefore, flavonoids have shown a positive effect on obesity and reduce its complications [137].

#### 3.1.3. Betalains

Betalains are water-soluble and nitrogen-containing pigments, divided into betacyanins and betaxanthins [138]. Normally, they are widely used as colorants. Diverse studies have indicated that betanin has antioxidant, anti-inflammatory [139], hepatoprotective [140], anticancer [141], and anti-diabetes activities [142].

#### 3.1.4. Carotenoids

Carotenoids have many effects on obesity, such as restricting the adipogenesis and hypertrophy of adipocytes [143]. According to Mounien et al. [144], carotenoids downregulate gene expression in adipocytes through NF-κB and MAPK, or via the transcription factors implicated in detoxification, such as aryl hydrocarbon receptor (AhR), nuclear factor erythroid-2-related factor 2 (NRF2), or Pregnane X receptor (PXR). In addition, they have an inhibitory effect on adipocyte differentiation, anti-adipogenic effects via the regulation of adipogenic transcription factors, such as C/EBPα (CCAAT/Enhancer-binding Protein α) and PPAR-γ (Peroxisome proliferator-Activated receptor), reducing LPS-mediated induction of TNF-α in macrophages via NF-κB and JNK, and attenuating macrophage infiltration [145], among others. The mechanisms of the action of specific bioactive compounds on animal models of obesity are summarized in Table 3.

### 3.2. Clinical Evidence

Clinical evidence regarding the effect of bioactive compounds to treat obesity and its comorbidities is limited compared with animal models of obesity, and clinical results are not conclusive. Although the evidence suggests that bioactive compounds are not effective for weight loss in humans, the anti-inflammatory response in the obesogenic state is still a field of research.

Clinical studies with a combination of bioactive compounds showed controversial results. In a pilot study, the effectiveness of dietary herbal supplements of rhubarb, ginger, astragalus, red sage, and turmeric was found to reduce food intake and cause weight loss in women with a 700 kcal/day diet. After 8 weeks, no changes in weight were observed [172]. The consumption of two cups of strawberry drinks daily by women with metabolic syndrome after 4 weeks reduced the levels of oxidized LDL without changes in CRP and adiponectin [173]. The acute consumption of 250 mL of Hibiscus sabdariffa calyces (HSC) extract, which is rich in polyphenols, for two weeks was proven in men with cardiovascular disease to increase the flow-mediated dilatation of the branchial artery. Although Gallic acid, 4-O-methylgallic acid, 3-O-methylgallic acid, and hippuric acid reached a maximum plasma concentration at 1 to 2 h post-consumption of the extract, changes in other clinical parameters and the CPR levels were not observable [174]. However, the chronic consumption of a combination of bioactive compounds (epigallocatechin gallate, capsaicin, piperine, and L-carnitine) for 8 weeks in overweight subjects showed diminished HOMA-IR, leptin/adiponectin ratio, LDL, ghrelin, and CRP [175]. The evidence suggests that some combinations could reduce their protective effects in clinical trials.

In trials with a single compound, the protective effect seems to be more evident. In a double-blind, randomized trial controlled with a placebo, 22 subjects with T2D received 180 mg of ellagic acid per day for 8 weeks and 22 subjects with T2D received a placebo. At the end of the study, fasting plasma glucose, insulin, HOMAIR, and Fetuin A were reduced and serum sirtuin1 was increased by the treatment of ellagic acid [176].

In a pilot study with subjects with multiple sclerosis, the consumption of 800 mg epigallocatechin gallate and 60 mL of coconut oil decreased IL6 and the fat percentage [177]. In overweight women and those with obesity, the consumption of epigallocatechin-gallate and resveratrol (282 mg/d and 80 mg/d, respectively) for 12 weeks did not cause changes in adipocyte size and distribution, but caused changes in pathways related to adipogenesis (β-estradiol and Prolactin), the cell cycle and apoptosis were downregulated, as well as oxidative stress (nuclear factor and erythroid 2-like 2 (NRF2)) and inflammation (TNF-α) [178].

Resveratrol consumption (150 mg/day) has been proven in obese men in a randomized double-blind crossover study for 30 days. The effects of Resveratrol were: decreased intrahepatic lipid content, circulating glucose, triglycerides, HOMA index, systolic blood pressure, and inflammation markers (CRP). Although changes in the BMI were not observable, this trial suggests that resveratrol induced metabolic changes in obese humans mimicking calorie restriction [179].

Another factor to take into account is the possible interference with meals. For example, a daily intake of 25 mg of pure (-)-epicatechin (EPI) for two weeks does not reduce cardiometabolic risk factors in overweight and obese adults [180]. However, the consumption of a higher dose of 100 mg of EPI before meals for 4 weeks showed a significant reduction in the TG/HDL ratio and hsCRP [181].

Additionally, genetic factors could interfere with the observable effects. In a study (double-blind, placebo-controlled cross-over trial with 6-week treatment periods separated by a 5-week washout period) with ninety-three overweight or obese adults with metabolic syndrome, the effect of 150 mg of quercetin was evaluated. The consumption of quercetin reduced the systolic blood pressure and plasma-oxidized LDL without changes in serum TNF-alpha and CRP [182]. This could be explained by a genetic predisposition; Egert and colleagues reported that the reduction of TNF-alpha was dependent on the apolipoprotein E genotype [183]. Later, the Egert group studied the effect of the consumption of 162 mg/d quercetin in overweight-to-obese patients with pre- and stage-1 hypertension, without changes in systemic and adipose tissue inflammation after 6 weeks of treatment [184].

The use of curcumin in clinical trials has been widely evaluated. The effects of curcumin in a 6-month randomized, double-blind, and placebo-controlled clinical trial with subjects diagnosed with type-2 diabetes showed a positive effect on the reduction of the pulse wave velocity and leptin with increased levels of serum adiponectin [185]. Ganjali and collages conducted a randomized, crossover, and controlled trial in obese individuals who consumed 1 g of curcumin daily for 4 weeks. They observed a significant reduction in IL-1β, IL-4, and VEGF with curcumin consumption [186]. In a study on overweight girls who consumed 500 mg of curcumin per day for 10 weeks, a reduction in serum IL-6 and CRP was observed [187]. However, in another study, the consumption of 1 g of a phosphatidylcholine complex of curcumin in individuals with metabolic syndrome resulted in no changes in the BMI and clinical parameters [188]. Additionally, the use of resveratrol in combination with curcumin has no impact on the postprandial inflammatory markers of obese individuals in an acute intervention [189], and the use of curcumin alone or with fish oil in older overweight adults and those with obesity did not result in additional benefits to the fish oil alone, which improved dyslipidemia [190].

The use of a network platform has been useful to evaluate the synergistic mechanism of Sanghuang–Danshen (SD) phytochemicals in the homeostatic protection against high-fat-induced vascular dysfunction in healthy subjects. The acute consumption of 600 and 900 mg of SD phytochemicals had synergistic effects and fumaric acid, cryptotanshinone, and ellagic acid would exert a synergistic influence on vascular health by regulating adhesion molecule production [191]. Therefore, the use of new bioinformatic tools could be useful for understanding the interactions of bioactive compounds and their potential effects.

### 3.3. Antioxidant Fiber Dietary

There is a diversity of plant and by-products that have been considered as potential sources of dietary fiber and bioactive compounds, in such a way that it has been called “antioxidant dietary fiber” (ADF) and is defined as a product with a content of natural antioxidants associated with the fiber matrix [192], which is characterized by the combined beneficial properties of both dietary fiber and antioxidants [193] and could be considered as a bioactive compound.

Dietary fiber maintains the functional integrity of the gastrointestinal tract, improves constipation, improves cardiovascular diseases and diabetes, and reduces the risk of developing cancer [169,193,194,195,196]. The action mechanisms that develop the fiber depend on the dietary fiber type. A diversity of studies on animals has demonstrated the effect of consuming dietary for obesity control, as well as to decrease gastric bloating and promote satiety, through the interaction between fiber (soluble and insoluble) and its contact with water to increase its viscosity, which will depend on different factors, such as the structure of the fiber, and the chemical composition, concentration, and molecular weight of the dietary fiber. Additionally, ADF decreases caloric intake, aids in weight reduction (fat), induces changes in body fat distribution, decreases the fatty tissues, and even inhibits glucose absorption, and high cholesterol and triglycerides levels [195,197,198,199,200,201,202,203,204,205].

Zou et al. [206] reported that insoluble fiber induces the expression of IL-22, which in involved in reducing the attack on the microbiota due to the building of the epithelium by the regeneration of crypts and the expression of antimicrobials that protect against a series of inflammatory processes, such as those produced by obesity [206]. The results obtained by Sanchez et al. showed that the intake of soluble fiber may enhance the pro-inflammatory state characterized in obesity [207], as fiber could protect against the oxidative stress characterized by this pathology. On the other hand, Ma. et al. [208] proved that soluble and insoluble fiber are related to low CRP concentrations, as fiber decreases lipid oxidation and, therefore, reduces inflammation. In addition, another study reported that fiber consumption decreases inflammatory markers due to reduced LPS production and improves gut permeability [209].

## 4. Conclusions

The modification of lifestyle is suggested as a treatment for obesity control, mainly including the consumption of natural foods, which could help to improve health due to the content of bioactive compounds, such as flavonoids, phenolic acids, or dietary fiber. In addition, other relevant aspects, such as the bioavailability, metabolic pathways, and action mechanics, of the resultant metabolites of bioactive food compounds are important aspects that reduce obesity and its related diseases. However, more research is required to justify the use, efficacy, and safety of foods with bioactive compounds. Additionally, clinical validation is necessary with finality to implement correct treatment strategies.

## Figures and Tables

**Figure 1 foods-11-01232-f001:**
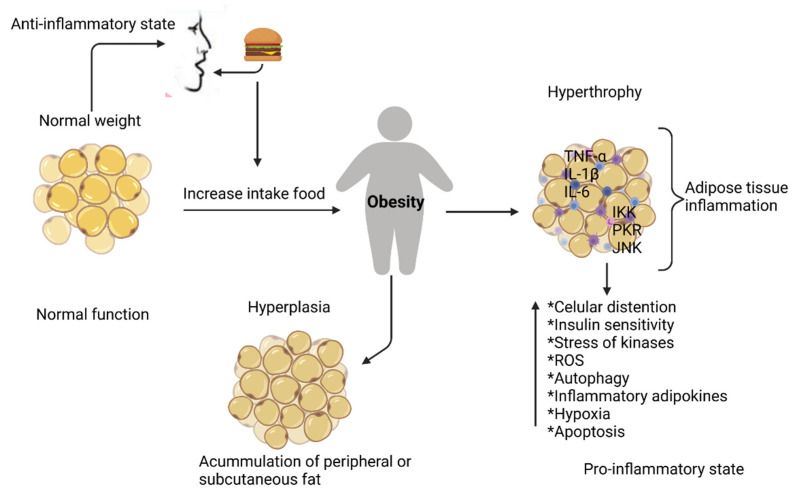
Adipose tissue inflammation. Excessive growth of adipose tissue in obesity induces the production of pro-inflammatory cytokines that activate protein kinase pathways, at the same time stimulating macrophage infiltration and a change in the phenotype of M2-type macrophages to proinflammatory M1, leading to an inflammatory state with consequences locally and systemically. Tumor necrosis factor-alpha (TNF-α), interleukin-1b (IL-1β), interleukin-6 (IL-6), N-terminal c-JUN (JNK), nuclear factor-kappa kinase inhibitor β (IKK), protein kinase R (PKR). Created with BioRender.com.

**Figure 2 foods-11-01232-f002:**
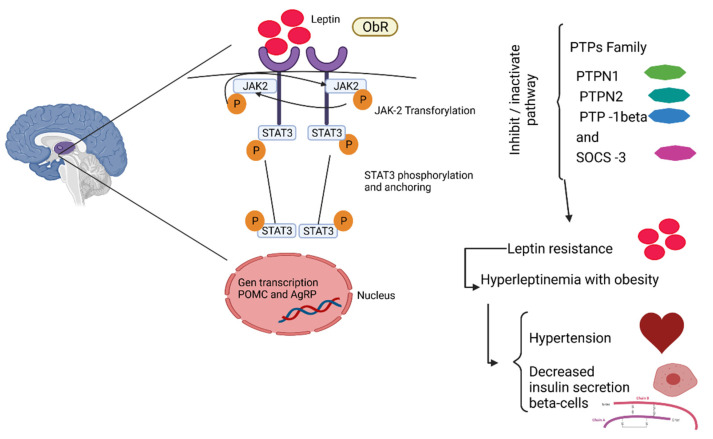
Leptin pathway. Leptin binds to the ObR receptor and JAk-2 transphosphorylation occurs, translocating the phosphate groups, giving rise to the anchoring and phosphorylation of STAT3. These STA3 travel to the nucleus, where the transcription of target genes, such as POMC (decreases hyperphagia) and AgRP (increases food intake), takes place. This pathway can be inactivated by the interaction of tyrosine-protein phosphatase 3 (PTP3) and suppressor of cytokine signaling 3 (SOCS3), causing resistance to leptin, resulting in hyperleptinemia, which leads to cardiovascular problems, such as hypertension, as well as causing a decrease in insulin secretion in β cells. Created with BioRender.com.

**Table 3 foods-11-01232-t003:** Molecular mechanisms of bioactive compounds on animal models of obesity.

Bioactive Compounds	In Vivo	Mechanisms of Action	Toxicity	Author
Phenol acids				
Caffeic acid	C57BL/6 mice with diet HFD	The mechanism focuses on an increase of the phosphorylation of AMP-activated protein kinase and decreasing acetyl carboxylase, a downstream target of AMP-activated-protein kinase (AMPK).	No maternal toxicity	[145]
Ellagic acid	High-fat diet-induced obesity SD rats.	Decreases the mRNA expression of Zfp423 and Aldh1a1 (responsibilities of WAT plasticity) and increases the mRNA expression of the brown adipocyte, as well as markers UCP1, PRDM16, Cidea, PGC1α, and Ppar-α; and beige markers, including CD137and TMEM26. It also elevates the expression of UPC1 in iWAT (specific protein of brown adipocyte).	No-observed-effect level 3011 mg/kg bw/day (males) No-observed-effect level 3254 mg/kg bw/day and 778 mg/kg (females) (rats)	[13,146]
Gallic acid	Mice (Swiss) model fed with high-fat diet	Induces an increase in SIRT1 and PGC1-α, might be responsible for thermogenesis activation under a high-fat diet.	Non-toxic >100 mg /L	[12,147]
*p*-Coumaric acid	Mouse model of high-fat diet-induced obesity	The mechanism of the action on obesity is mediated by the mTORC1-RPS6 pathway, regulating the Ucp1, HSL, and GUT-4 proteins	Low toxicity 2850 mg/kg bodyweight (mice)	[14,148]
Vanillic acid	High-fat diet (HFD)-induced obese mice and genetically obese db/db mice	The mechanism of action is due to the increase in the cellular NAD levels, and AMPK activates the NAD-dependent deacetylase SIRT1, which results in the deacetylation or activation of PGC1 and, therefore, a thermogenic effect.	1000 mg/kg b.w (rats)	[149,150]
Flavonoids				
Capsaicin	Mouse (Adult male WT and TRPV1−/− (B6.129X1Trpv1 tm1Jul/J) model of HFD-induced obesity.	Intracellular Ca2+ rises via TRPV1 channels stimulated by CAP, activating CaMKII/AMPK, which phosphorylates and activates SIRT-1. This causes the deacetylation of PPAR-γ and PRDM-16 and facilitates their interaction to promote the browning of WAT (white adipose tissue).	Oral LD50 118.8 mg/kg for males and 97.4 mg/kg for females (mice) Male rats—161.2 mg/kg, and female rats—148.1 mg/kg	[151,152]
Anthocyanins	Male C57BL/6J mice fed a modified AIN-93M control diet containing high fat/high cholesterol	Inhibition of IKKε expression in adipose tissue occurs. Prevents the action of macrophage infiltration by attenuating the action of IKKε in energy preservation.	No toxic effects of anthocyanins identified 20 mg/kg/d mice; >3 g/d guinea pigs and rats; >2.4% body weight in beagle dogs and 9 g/kg/d in rats, mice, and rabbits	[153,154]
Pterostilbene	Zucker rats (fa/fa) model of genetic obesity	Present effect thermogenic and oxidative capacity of brown adipose tissue, due to increase of gene expression of Ucp1, peroxisome proliferator-activated receptor γ co-activator 1 α (Pgc-1α), carnitine palmitoyl transferase 1b (Cpt1b), nuclear respiratory factor 1 (Nfr1), and cyclooxygenase-2 (Cox2); PPARα, PGC-1α, p38 mitogen-activated protein kinase (p38 MAPK), UCP1 and glucose transporter (GLUT4); and enzyme activity of CPT 1b and citrate synthase (CS) were assessed in interscapular brown adipose tissue.	No significant toxic effects	[11,155]
Resveratrol	High-fat diet (HFD)-induced adipogenesis and inflammation in the epididymal fat tissues of mice C57BL/6J.	There are changes in the GalR1, GalR2, PKCd, and p-ERK protein expressions, with subsequent changes in the Cyc-D and E2F1 expressions, on galanin-mediated adipogenesis cascades in the epididymal adipose tissue. Decrease adipogenic transcription factors (PPARg2, C/EBPa, SREBP-1c, and LXR) and their target genes (FAS, LPL, aP2, and leptin) were suppressed. TLR4 uses MyD88-dependent and MyD88-independent pathways, whereas TLR2 signals only in the MyD88-dependent manner. The MyD88-dependent pathway uses TRAF6 and IRF5, leading to its nuclear translocation and cooperation with NF-kB. The MyD88-independent pathway uses TRIF in activating NF-kB in either a TRAF6-dependent or TRAF6-independent mechanism. TRIF associates with TBK1 and IKKi, which in turn leads p-IRF3. Resveratrol limits changes in the expression of TLR2, TLR4, and downstream molecules (MyD88, Tirap, TRIF, TRAF6, IRF5, p-IRF3, and NF-kB), along with the subsequent changes in the cytokines (TNFα, IFNα, IFNβ, and IL-6) implicated in the TLR2/4-mediated pro-inflammatory signaling cascades on adipose tissue	No toxic effect in humans	[10,156]
Curcumin	Mice C57BL/6 fed a high fat diet	There is a suppression of acetyl CoA conversion to malonyl CoA. Lower levels of malonyl CoA increase CPT-1 expression, promoting fatty acid oxidation. The phosphorylated AMPK also suppresses the expression of GPAT-1, which results in reduced fatty acid esterification. The phosphorylated AMPK inhibits PPAR-γ and C/EBP-α transcription factors.	No toxicity from curcumin	[157,158]
Quercetin	Diet-induced obese (DIO) ICR mouse	Blocked protein levels of the key adipogenic factors C/EBPβ, C/EBPα, PPARγ, and FABP4, and the TG-synthesis enzymes lipin1, DGAT1, and LPAAT. Inhibited MAPK, ERK1/2, JNK, and p38MAPK, and MCP-1 and TNF-α in adipocytes and macrophages	285–3000 mg/kg toxicity present	[159,160]
Apigenin	High-fat diet (HFD)-induced obese C57BL/6 (C57) mice	Apigenin binds to non-phosphorylated STAT3, reduces STAT3 phosphorylation and transcriptional activity in visceral adipose tissue, and consequently reduces the expression of the STAT3 target gene cluster of differentiation 36 (CD36). The reduced CD36 expression in adipocytes reduces the expression of peroxisome proliferator-activated receptor-gamma (PPAR-γ) which is the critical nuclear factor in adipogenesis.	300 mg/kg (mice) No toxicity	[161,162]
Scutellarein	Mouse model of obesity induced by high-fat diet (HFD) feeding.	There is suppression of the expression of cytokine genes TNF-α, IL-6, IL-1β, ICAM-1, VCAM-1, and NF-κB.	Minimally toxic or non-toxic in rodents	[163,164]
Luteolin	C57BL/6J mice model of DIO (diet-induced obesity: high-fat diet)	It is modulated the TLR signaling pathway on pro-inflammatory response. There is a decrease in EMR1 and CCL7, which impacts adipose tissue, increases lipolysis and the TCA cycle, reduces the pro-inflammatory response, adipokine dysregulation, adipocyte macrophage infiltration and accumulation, fibrosis, pancreatic β cell dysfunction, hepatic lipotoxicity, insulin resistance, and chronic inflammation. Another mechanism of action is the interaction in the AMPK/PCG1α. Elevates the expressions of thermogenic genes and the activities of AMPK/PCG1a signaling molecules.	No adverse effect or toxicity	[165,166,167]
Chlorogenic acid Caffeine	ICR mice with high-fat diet	Increases AMPK phosphorylation and p-AMPK up-regulates the expression of ATGL and HSL, promoting the hydrolysis of triglycerides and the release of FA. Elevates ACO expression by the activation of AMPK (accelerated β-oxidation). Down-regulation of LXR-α and increase in p-AMPK restrain the expression of SPEBP1c, thereby down-regulating the expression of SCD1 and FAS to inhibit lipid synthesis and regulate lipid metabolism.		[168]
Catechin, Picatechin, Procyanidins	High-fat diet-fed C57BL/6 mice	Activated AMPK-α also induces the expression of UCPs and PGC-1a, which are involved in energy expenditure and thermogenesis		[169]
Cyanidin-3 O galactoside	Mice (C57BL/6) model with high-fat diet-induced obesity	Related to adipogenesis-related transcription factors (C/EBPs, PPAR-γ, and SREBP-1c) and coactivators (PGC-1α), and the down-regulation of specific adipogenesis-related genes affected by these transcription factors.		[170,171]
Other compounds				
Betacyanins	High-fat diet (HFD)-induced obese mice	Reduces HFD-induced body weight gain, and ameliorates adipose tissue hypertrophy, hepatosteatosis, glucose intolerance, and insulin resistance. Increases the expression levels of lipid metabolism-related genes (AdipoR2, Cpt1a, Cpt1b, Acox1, PPAR-γ, Insig1, and Insig2) and FGF21-related genes (β-Klotho and FGFR1/2), and decreases the expression level of Fads2, Fas, and FGF21		[171]

## Data Availability

Not applicable.

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
