# Peer review of "Role of Bioactive Compounds in Obesity: Metabolic Mechanism Focused on Inflammation"

_foods, 2022, doi:10.3390/foods11091232_

Round 1
Reviewer 1 Report
This review aims to provide a comprehensive overview on recent studies about the possible role and effect of specific bioactive compounds on weight management, and obesity consequences.
This review is a systematic review that focuses on the latest data on the molecular mechanisms of polyphenols and dietary fiber based on publications from Scopus, Science Direct, PubMed, and other databases, using keywords and based on recent in vitro and in vivo studies .
There are sufficient references to significant publications on the topic of the work for the last 10 years.
The literature review ends with scientifically substantiated conclusions. The abstract adequately reflects the main points of the work.
Author Response
"Please see the attachment."

Reviewer 2 Report
The manuscript is overall well written. However, I have a few minor concerns.
- Why only two diseases are discussed in section 2. Obesity can be linked with multiple diseases related to metabolism. The authors must incorporate a few more subsections.
- In each disease section, the objective role of obesity in disease progression is poorly described. The authors need to incorporate a few more recent publications describing the molecular involvement.
- The major classification of bioactive compounds gives an overall idea. However, specifying a few compounds with diverse applications will be more helpful for the readers.
- The toxicities related to few bioactive compounds are not discussed.
- The specificity of the bioactive compounds as individual or multi-disease targeting modalities must be discussed and a separate table regarding this can be included.
- The clinical and pre-clinical relevance and applications are not discussed.
- I find the figures to be more generalized. Can the authors put some mechanistic view of pathways associated with obesity and disease progression in figures?
- The language must be double checked.
Author Response
"Please see the attachment."

Reviewer 3 Report
The review "Role of bioactive compounds in obesity: metabolic mechanism 2
focused on inflammation" is well written. however the author should include the a brief mechanism of some bio-actives and their effect on obesity. References from some recent literature should be included.
Author Response
"Please see the attachment.

Reviewer 4 Report
The authors review the metabolic mechanism regarding different bioactive compounds in obesity, mainly focused in the obesity-derived inflammation. In general, the manuscript is very interesting based on the interesting role of natural products with both nutritional and pharmacological characteristics, such as polyphenols. Also, the authors included current in vitro and in vivo data/studies, which also improve the quality of this review
From my point of view, this manuscript is highly interesting for the readers. However, English language should be improved..
Author Response
"Please see the attachment.
